# Understanding Teacher Educators' Perceptions and Practices about ICT Integration in Teacher Education Program

**Ayubu Ismail Ngao** [1,2] 🆔, **Guoyuan Sang** [1,]* 🆔 **and Jimmy Ezekiel Kihwele** [3] 🆔

1    Faculty of Education, Beijing Normal University, Beijing 100875, China
2    Department of Educational Foundations and Management, Mkwawa University College of Education,
     Iringa P.O. Box 2513, Tanzania
3    Department of Education Foundations and Teaching Management, Mzumbe University,
     Morogoro P.O. Box 1, Tanzania
*    Correspondence: guoyuan.sang@bnu.edu.cn

**Abstract:** This study explored the perceptions and practices of teacher educators in integrating information and communication Technology (ICT) in teacher education programs. The study adopted a phenomenological design under the qualitative research approach that included eighteen selected participants from a teacher education university college. Data collection employed semi-structured interviews, observations, and documentary reviews. The authors adopted the Braun and Clarke (2006) thematic analysis model for data analysis. The findings showed that while young and inexperienced teachers showed readiness to use ICT, some teacher educators do not understand the logic behind using technology and hence question the rationale for applying it to their teaching. At the same time, equipment challenges, large teaching burdens, and time limits were the critical barriers to integration. Again, the findings revealed that teacher educators use different software and learning platforms, use social media, gather online information, and access learning materials through journal subscriptions to enhance preservice teachers' learning. Thus, integrating ICT during teacher training is paramount, and teacher educators should be assisted and encouraged to develop positive attitudes in learning and to apply ICT in their teaching practices. Concomitantly, equipping preservice teachers with ICT-based pedagogical skills, not only through specialized ICT courses, but also through observing how teacher educators use it, has a significant impact on transforming teaching practices in their future classrooms.

**Keywords:** information and communication technology (ICT); teacher educator; teacher education

## 1. Introduction

Integrating Information and Communication Technology (ICT) in education provides a magical power of sustaining teaching and learning beyond unexpected interruptions. The enormous flow of information and use of technology emanated in all fields worldwide [1]. Studies have shown the significance of using ICT in teaching and persuading the vital essence of a deep personal understanding of ICT [2–5]. This significance is not limited to effective usage or its reason and application during instructional practices. It is a fact that ICT is prevalently adopted in the educational field for various reasons, such as making instructional practices successful for learners and instructors [6], increasing students' motivation in learning [7], refining learners' collaboration and engagement with the lesson [8], and improving the quality of teaching [6]. These facts make technology in education a vital element in enhancing learning in these centuries.

Although technology in education, particularly in the classroom environment, is significant, its application is too scarce in some contexts. Various reports inform governments to increase the accessibility of digital technologies to teachers and students to improve the learning process [9,10]. These technologies are not limited to the convenience of internet connections and different technological devices. However, teachers at the center need to

possess enough skills to utilize those devices, such as digital media, cable networks, TV, and radio, as well as social media like WeChat, WhatsApp, Facebook, LinkedIn etc. Gulbahar [11] claimed that equipping schools with these technological devices do not improve teaching quality when teachers lack the skills to effectively utilize them. Here we can conclude that without teacher competence in the effective utilization of these technologies, it will prevent teachers from attaining the intended goals [4]

Moreover, the full integration of technology in teacher education programs provides the basis for effective and critical instructional practices. Teachers can enhance higher-order thinking through student's engagement in complex tasks through collaboration [12] (p. 519), strengthening teamwork [7], knowledge sharing and critical thinking [13,14], holistic learning, and independent research, as well as enhancing teachers' and students' motivation to collaborate [8,15].

Studies have disclosed deficient use of ICT in teacher education programs [13,16–18]. This low use of ICT has resulted in the incompetence of preservice teachers in the aspect of the analysis of the problem, evaluations, synthetization, solving the problem, and the effective drawing of attainable conclusions about their profession. Regarding these impacts highlighted by different scholars, this study aimed to understand or study teacher educators' perceptions and practices on integrating ICT in their teaching and learning process. With this matter, the study focused on the following line of inquiries. (1) What are the teacher educators' perceptions of integrating ICT in their teaching and learning process? (2) How do teacher educators integrate ICT during teacher training?

## 2. Literature Review

### 2.1. Theoretical Perspective

This study espoused the Unified Theory of Acceptance and Use of Technology (UTAUT), as adopted by the scholar Venkatesh [19] (p. 447). Fred Davis and Richard Bagozzi developed this theory in 1989. The vastly used revision is Technology Acceptance Model (TAM) and (UTAUT), as Davis highlights [20]. Developing the unified framework for empathetic technology acceptance combined eight different models' main prominent features. The framework allows the individuals to comprehend cognitive processors between the users and show how they respond to adopting and integrating the new piece of technology within their life. Nevertheless, in this context, the piece of technology denotes computer hardware, software, and online learning communities.

Implication of the Theory to the Study

The theory proposed that teacher educators should be optimistic about using new technology in their teaching. At the same time, tracking the improvement gained through new technologies can help assess the feedback and the productivity rate. Studies on preservice teachers and teacher educators, respectively, found they have a diverse understanding of ICT use in their teaching and learning, while others expressed discontent on the use of technology [21,22]. In this regard, the prospective user's perception of whether or not using a particular application system will enhance their performance within an organization is important. Consequently, understanding their perception will determine the nature of the technique to engage and assist in plans to transform their private theories to benefit from ICT use.

Venkatesh [19] highlighted that the calmest helpful technology could be leisurely through user feedback. Meanwhile, the literature highlighted the lack of support as the limitation to technology integration. The theory believes in the strong support for effective use of technology. The theory anticipated that technology applied during instructional practices should be learner-friendly, implying that there should be no difficulties during the integration process. In contrast, the theory contends that the educators' age, awareness of technology use, behavioral intentions, facilitating conditions, and social issues, influence educators' actual usage of technology in their teaching and learning. Similarly, the theory suggests that the institutional, technical, and physical infrastructures require

backing from the prevailing technologies. This backing includes internet accessibility in the institution and teacher educators' knowledge on how to use the existing technologies for the advancement of the preservice teacher.

In a nutshell, UTAUT is purposely chosen for this study because it calls for a holistic understanding of the other variables affecting teacher educators' perceptions and practices on integrating ICT during teaching and learning. However, the factors identified in this theory seem relevant to teacher educators when assessing their practices. Ultimately, the current study aimed to understand their perceptions and actual practices regarding ICT integration.

### 2.2. Overview of ICT Integration in Education

Globally, education systems are under increasing pressure to use advanced technologies to teach learners the essential competencies desired in these epochs [4]. For instance, the U.S government dispensed the first Nationwide Educational Technology Plan; the plan intended to prepare learners for the 21st century through effective training and provide auxiliary educators and students with computer competencies and internet accessibility [12]. Similarly, EU Science Hub 2016 espoused a new and inclusive skills program for Europe, which intended to ensure that all universities, colleges, and schools across Europe fully integrate technology during their instructional practices [7]. Meanwhile, many Asian countries have confidence in ICT, which positively impacts students and builds instructors' capacity to develop the eminence of the learning process [23,24]. For example, in 2008, Massive Open Online Courses (MOOC) appeared to be the most used platform across China to support distance learning [25]. The platform promotes collaboration among the users with essential resources which enhances their motivation and ambition to complete the program successfully. Additionally, the platform transforms teacher professional development by providing better access to quality professional learning opportunities for teachers in remote, rural, and urban areas [26]. Educause [27] highlights other benefits, such as no limits to admission, anyone being able to join the program, and cost-effectiveness; moreover, the course is well-structured around attained learning goals. However, the teacher's role is significant in improving student's participation and interaction during learning processes [25].

### 2.3. Strategies Adopted for Integrating ICT in Teacher Training Program

In addressing the current issue, European countries devoted their exertion to conniving Teacher Professional Development (TPD) programs that support the entire process of evolving competencies of technology use among teacher educators in teacher education universities and secondary schools [7]. Furthermore, to appropriately advance the required competencies for teacher educators, they used the teachers' ICT competence model established by different world organizations [28–30]. The main target of this model was to provide teacher educators with, at slightest, a basic set of essential competencies that allow teacher educators to integrate ICT when teaching, help learners to advance their learning, and rally their professional daily responsibilities. The implication is that studying different competency frameworks from diverse viewpoints helps to promote teacher educators' abilities to integrate ICT. Similarly, UNESCO [10] carried out research on transversal competencies in nine different countries in Asia. The research results come with different strategies to improve technology integration in the teaching and learning process, and include the assertion that teachers' roles should change from spreaders of knowledge to facilitators of knowledge. Besides this change, teachers must adopt student-centered learning strategies emphasizing collective or collaborative team-based learning.

### 2.4. Challenges of ICT Integration in Teacher Education Program

Many countries stress techniques and infrastructure to sustain the use of ICT in their education system. Supporting ICT use is not a severe problem for some countries in Europe and Asia [23]. The technological learning infrastructures are extensively accessible, though the full potential of technology is not subjugated in school education [31]. Among

many identified reasons behind this was a lack of teachers' competencies or familiarization with technology use [32]. This observation shows the importance of imparting to teacher educators the required competencies that will act as a catalyst to ease the teaching process. Additionally, Teng noted the same barrier as a Chinese scholar who testified that teacher educators lack adequate professional training, which affects technology use during teacher training [23]. Some teachers are uneasy with technologies, have accessibility issues, and feel secluded from the digital-age teaching communities. Under these situations, teacher education programs at universities, colleges, and schools must pay extra courtesy to integrate ICT with the pedagogy, as suggested in the TPACK approach. His framework proposed teaching in the educational arena should go from technology-centered training to technologically integrated teaching.

*2.5. ICT in Teacher Education Universities in Africa*

UNESCO organized a new high-priority Ingenuity on Teacher Training in Sub-Saharan Africa (TTISSA) in 2006. The initiative intended to refine the eminence the teacher training around 46 selected countries in SSA [33]. The initiative showed that many universities in African countries conveyed unadorned technology constrictions such as connected internet and computer hardware [30]. Another project was the Continental Educational Strategies for Africa (2016–2025) under CESA [34]. Within the project, they reached a unified understanding of technology integration in educational learning, which extended from STEM subjects to all other subjects. Relying on the educational report from UNESCO, the eminence of preservice training in SSA does not accentuate contemporary issues and relies more heavily on theories rather than practices [35]. These issues are not limited to critical constructive learning, student-centered approaches, and the full integration of new technologies into instructional practices.

For instance, in the context of technology use in teacher training in Nigeria, there is an inability of different teacher training to use technology in teaching and learning, which negatively affected the quality of teachers produced, who are thus less competitive in the market [36,37]. With this situation, they developed special professional training for teacher educators and restructured the entire teacher education program with technology elements [28]. A similar situation is viewed differently in teacher training in South Africa, where the government has comprehended the significance of technology integration in teacher training [33]. All teacher training institutions were required to create a conducive environment for the full integration of technology for the entire curriculum of teacher education programs [38]. However, studies cited the program was more urban-centered while other training institutions lack the basic technological devices [33,38,39]. Lacking essential devices affects teacher educators who lack enough competencies to use technological devices and preservice teachers, especially in their future classroom practices.

In Tanzania, teacher training is under teacher education colleges and universities. The Tanzania Commission for Universities (TCU) is responsible for ensuring the quality of training and education delivery. The Ministry of Education Science and Technology prepares undergraduates and postgraduates. The board at the ministry level is responsible for administering and managing teacher training programs across the country. Major educational reforms, such as policies, curriculum reforms, and pedagogical and assessment reforms, have been carried out to ensure technology is well integrated during teacher training [40–42]. Despite different reforms in teacher education programs, the integration of ICT in teaching and learning has become a critical issue which affects the quality of teachers produced. For instance, out of many other obstacles, low integration of technology during instructional practices is a critical issue in teacher education programs in Tanzania [17,18,43]. This low ICT integration prevents the country from getting qualified and competitive teachers in this technological revolution era.

Additionally, various projects have been launched, from local and international levels, to ensure technology is fully integrated into teacher training. For instance, The Comprehensive Framework for Teachers (ICT-CFT), in collaboration with the International Society

for Technology in Education (ISTE), Microsoft, Intel, and other international organizations, was launched to assist technology integration in education [44]. The main barrier in implementing these projects is digital illiteracy among teacher educators and preservice teachers. At the same time, the Ministry of Education [45] showed initiatives toward technology integration in teacher training and highlighted:

"Teacher education programs would be oriented towards enabling preservice and in-service teachers to become multitalented teachers, capable of thinking critically and competent in integrating technology in learner-centered teaching situations".

The implementation of these initiatives strives to ensure the full integration of ICT during teacher education training. However, the reviewed studies showed teacher educators lack a theoretical and practical understanding of ICT use in their teaching and learning process; in this regard, this study aimed to understand teacher educators' perceptions and practices on integrating ICT during the teaching and learning process.

## 3. Methodology

This study describes the lived experience of teacher educators and heads of different units at the University College, such as library and ICT, concerning teacher educators integrating technology in their teaching practices and understanding their perceptions towards it. The study employed a phenomenological research design which helped to explore the phenomenon from the lived experience of the participants concerning the integration of ICT in teacher education programs [46]. The design involves extracting and analyzing lived experiences of several participants who participated in the study. The selection criteria for a study area included the institution's accreditation status, the nature of ownership, and geographical location.

### 3.1. Area of the Study

The area of study was Mkwawa University College of Education (MUCE), a constituent college of the University of Dar es Salaam in Tanzania. Established in 2005, the college has three faculties: the Faculty of Education, the Faculty of Humanities and Social Sciences, and the Faculty of Science. MUCE offers bachelor's degrees of education in arts and science subjects. The college uses Tanzania's curriculum to prepare teachers to teach at various levels, from primary to middle teacher training colleges. The training duration of the program is three (3) years. Apart from bachelor's degree programs, the college offers various postgraduate and master's degree programs in the education field.

### 3.2. Sample and Sample Size

The selected institution for this study had 85 staff members. Researchers used a purposive sampling technique to ensure a good representation of different teachers with different characteristics. Thus, 18 participants were selected to participate in this study. The data showed that a greater number of people who participated in this study were male (61.1%), which was greater than the expected 50%, and females represented 38.8% of participants, which is 11.2% less than the expected 50%. The participant's age distributions are as follows: 45 years and above (38.8%), 35–39 years (22.2%), 40–44 years (22.2%), and 30–34 (16.6%). Table 1 shows detailed information about the respondents.

Furthermore, respondents were academic staff with the following educational qualifications: PhDs represented 61.1%, followed by master's degrees, which represented 22.2%. Lastly, tutorial assistants with qualifications of a bachelor's degree represented 16.6% of participants. Furthermore, the data shows the working experience of academic staff of 11–15 years was the dominant category, represented by 44.4%, followed by 6–10 years with 33.3%, then 1–5 years with 16.6%. Lastly, greater than 16 years represented 5.5%.

**Table 1.** Characteristics of research participants.

| Code | Professional Title | Age (Yrs.) | Gender | Experience (Yrs.) | Specialization |
|---|---|---|---|---|---|
| TX1 | Senior Lecturer | 49 | M | 15 | Biology |
| TX2 | Lecturer | 42 | M | 12 | Geography |
| TX3 | Assistant Lecturer | 35 | F | 8 | Educational management |
| TX4 | Tutorial Assistant | 32 | M | 2 | Chemistry |
| TX5 | Lecturer | 56 | F | 24 | Educational Policy |
| TX6 | Lecturer | 40 | M | 12 | Physics |
| TX7 | Tutorial Assistant | 33 | F | 3 | History |
| TX8 | Senior Lecturer | 51 | M | 15 | Mathematics |
| TX9 | Senior Lecturer | 49 | F | 13 | Linguistics |
| TX10 | Lecturer | 41 | F | 10 | Development studies |
| TX11 | Assistant Lecturer | 36 | M | 6 | Educational Policy |
| TX12 | Tutorial Assistant | 34 | M | 4 | Psychology |
| TX13 | Lecturer | 48 | M | 13 | Education curriculum |
| TX14 | Lecturer | 45 | F | 15 | Linguistics |
| TX15 | Senior Lecturer | 49 | M | 14 | Mathematics |
| TX16 | Assistant Lecturer | 39 | F | 7 | Foreign Languages |
| TX17 | Lecturer | 41 | M | 10 | Educational psychology |
| TX18 | Assistant Lecturer | 39 | M | 6 | Educational Policy |

### 3.3. Data Collection Methods

This study's data collection methods involved in-depth interviews, documentary reviews, and observations to obtain adequate data. An interview is essential to deeply understand the phenomenon and participants' experiences [47]. Observations and documentary reviews helped to balance and verify the obtained information. The authors obtained all the relevant information needed to answer the research questions.

#### 3.3.1. Interviews

Researchers informed the participants of the study's objective and how the gathered information is for learning purposes. Researchers again assured the participants that they would adhere to all research ethics during the data collection, presentations, and reporting of the findings. Participation in this study was voluntary; a total of four participants quit the study in the early stage of data collection. As all participants were academic staff with various teaching responsibilities, they had to decide a convenient time and place for conducting the interview. The interview questions aimed at grasping the participants' understanding of the ICT integration in teacher education programs. For instance, how do you perceive the use of ICT in teaching and learning? How do you integrate ICT to enhance the teaching and learning process? What factors encourage or discourage the integration of ICT in your teaching? Is there any institutional support? Participants gave their consent to participate in face-to-face interviews and have the conversation recorded for further analysis procedures. As 18 participants were purposively selected to represent the whole population, Table 1 presents more information about the interviewees and the characteristics of the sample selected. The time for semi-structured interview ranged from 18–35 min.

#### 3.3.2. Documentary Review

Researchers employed a documentary review to verify the data collected from the interview. The review involved the following key documents from the university:

1. The teaching timetable to understand the number of sessions for the selected participants per week.
2. The university policy on integrating ICT in teaching.
3. The workload policy, which establishes the teaching workload for each participant.
4. A list of registered students in each course.

### 3.3.3. Observation

Researchers conducted observation in the classroom situation during the teaching and learning process to ascertain teacher educators' use of ICT in teaching. The observation checklist involved teacher educators' use of ICT facilities in the classroom, availability of ICT facilities within the classroom (i.e., projection screen, projectors, laptops etc.), and the number of students per class.

### 3.4. Data Analysis

An analysis of qualitative data follows the iterative process, which consists of six primary phases: familiarization with data, generating initial codes, searching for themes, reviewing themes, defining and naming themes, and producing the report [48,49] (see Table 2). The process involved sorting data communicating similar meanings and organizing them informatively to answer the research questions. Before starting the analysis, researchers read the data at least five times to understand it. The computer software assisted in managing the generated initial codes to ensure that coding is well inclusive, thorough, and systematic [50]. After that, the development of sub-categories and themes followed [48] (p. 82) and [50]. In this process, the authors reviewed the codes generated, identifying the areas of similarity, and merging the codes. Those miscellaneous themes with the codes that did not fit the merged themes formed new independent themes while discarding irrelevant codes and categories.

**Table 2.** A sample of the coding procedure.

| Transcription | Initial Codes | Categories | Themes |
|---|---|---|---|
| I can't use ICT because I Can't access them easily. | Lacking skills to use digital technologies | Digital illiteracy | Perceptions of ICT integration in teaching and learning |
| I always use a traditional approach to deliver. | Preference for traditional teaching methods without ICT | | |
| I teach a general course to all first-year students. Their total number is 2198. | Huge workload limits technology integration | Vast teaching load | |
| I have more than three teaching groups per week | Fixed timetable per week | | |

The adopted approach of interpretative, especially semi-structured interviews, and comparing them from observation and documentary review makes the presentations and discussions of teacher educators' perceptions more interesting. The presented findings concisely and precisely describe the phenomenon and the concluding interpretation. In reporting data analysis, the authors used quotations to illustrate and support the themes based on the study questions. At the same time, the authors applied the anonymity principle to hide the respondents' identities through the use of codes (see Table 1). The authors also employed triangulation to ensure the credibility of the findings [51]. All the presented findings were checked and revised by the selected participants.

### 4. Results

The study explored the perceptions and practices of teacher educators in integrating ICT into teacher education programs. Understanding teacher educators' attitudes and beliefs is central to determining technology's role and effectiveness during teaching and learning. Their attitudes and opinions on educational technology and pedagogy, in general, ultimately influence how teachers integrate technology into their practices. This section presents the key findings of this study in the form of themes and sub-themes, starting with perceptions and followed by practices.

*4.1. Perceptions on ICT Integration in Teaching and Learning*

The study's first objective was to explore TEs' perceptions of integrating ICT during preservice teachers' preparation. The findings revealed various factors relating to TE perceptions. The factors revealed are digital illiteracy, limited time per session, and the logic behind using ICT. Other factors are vast teaching load, equipment challenges, and disregarding pedagogical knowledge. The following sections present detailed descriptions of these factors.

4.1.1. Digital Illiteracy

The findings show that the majority perceive themselves as lacking the ability and skills to use digital technologies for different reasons. Many grew up without access to technologies, such as a personal computer and the internet, but students today are raised in an environment saturated with computer technology. These "digital natives" can intimidate teachers with little technological experience. Teachers feel they do not have the necessary competencies when using technology. TX5, TX1, and TX4 clearly stated the incompetency as they discoursed:

> *I feel less in control of my class, fail to use technology effectively and am unlikely to explore new possibilities that utilize technology when designing my classroom.* [TX5]

> *I lack the confidence to use ICT in my class. I always use a traditional approach to deliver.* [TX1]

> *I understand our students are digital natives. I can't use ICT because I Can't access them easily!* [TX4]

The findings from the classroom showed that, in some classrooms, teacher educators did not use ICT facilities already installed in the classroom, like projectors, to explain some mathematical concepts; instead, they opted for the whiteboard. This situation confirms the findings from interviews that little experience in technology might intimidate them from using ICT in teaching.

The research showed digital literacy has a more significant influence on the integration of ICT during the teaching and learning process. ICT illiteracy leads to many teachers sticking to traditional teaching methods that disregard the use of ICT. Teachers who are less fluent with technology maintain a feeling of control during teaching and will not have to prepare to face the challenges of instructing digital natives in a digital environment. If teacher educators lack enough knowledge and skills about technologies, it will automatically affect the integration process.

4.1.2. Limited Time per Session

At the same time, some of them argued about the time limit, which is only one hour per session. Limited time affected teacher educators to integrate ICT into their teaching and learning process. For example, TX2 had the following to say "...the problem comes when the time given is only sixty minutes per session and the content to be covered is wider than the time given" [TX2].

In one session, a researcher observed that the teacher educator spent nearly ten (10) minutes trying to connect a projector for lesson presentation. In the context where the session has only sixty minutes, teacher educators can easily decide not to use ICT facilities when they know they might face technical problems connecting the facilities.

Most respondents reported time limits per session, limiting them from reaching their objectives. They argued that using the traditional approach is time effective, but they need to assign students different tasks so that students can feel involved.

TX4 confirmed the findings as he said, "Even though the classroom is less supportive, I always assign them with many tasks and assignments for group discussion" [TX4].

Their role during the seminar sessions is to facilitate discussion. Finally, teachers realized that using ICT during the learning process is time-consuming and needs com-

prehensive preparation before the session. Additionally, it helps to increase classroom participation, and students can easily adapt to their future teaching.

### 4.1.3. The Logic behind the Use of ICT

Respondents showed different opinions on the logic behind using ICT in their teaching and learning. On this question, TX2, who has more than 15 years of teaching experience at the university, said, "I do not understand the logic behind using ICT in teaching and learning. From my understanding, student-teacher do not need to be equipped with ICT since our students' environment does not have those ICT facilities" [TX2].

However, these findings contradict the teaching environments in the same institution. The observation noted that some ICT facilities are present in classes, and some teacher educators have used them to facilitate teaching and learning.

The above response is similar to TX5, who stated that "Equipping student-teacher with local teaching methods will help them more than ICT. Teachers can use local teaching aids and facilities to create a conducive environment for learning" [TX6].

This feedback proved that some teacher educators negatively perceive the integration of ICT in the teaching and learning process. In this situation, student-teachers lack mentorship from their respective teacher educators, which affects their future classroom teaching. Additionally, those interested in using technology in their teaching lack support from their colleagues.

### 4.1.4. Vast Teaching Load

Teachers' vast teaching loads limit them from integrating technology in their teaching and learning. Due to the increase in enrolment at the institution, the lecturer–student ratio is huge to the extent that lecturers always have to divide student-teachers into small groups. During the interview, one of the respondents said:

> *Using technologies in my teaching will be very difficult for me. I teach a general course to all first-year students. The total number is 2198. We group these students into small manageable groups. I cannot manage to use ICT all the time due to a large number of students per session, making preparation and presentation time-consuming.* [TX8]

The review of the registered students revealed all 2198 first-year students studied one crosscutting course. The university divided these students into three groups, each with one weekly lecture. The observation in the classroom further revealed that teacher educators hardly used ICT facilities since the big lecture hall and the projector could not serve the purpose.

Under the same question, some complained that they have more than three teaching groups per week, and each group has more than 500 students. The use of technologies will be time-consuming and limit the coverage of the content. However, from the above complaints, they suggested the following: increasing the number of academic staff to reduce teaching loads, seminars and workshops for all academic staff on how to use technology in big sessions, and increasing the time per session from one hour to at least two hours.

### 4.1.5. The Challenge of Equipment Accessibility

Inadequacy of ICT-related devices has a significant influence on teacher educators' perceptions. The observation findings indicate that some teacher educators fail to integrate ICT during teaching and learning due to limited access to ICT facilities to support integration. We categorize these challenges into two main categories: (i) the individual level and (ii) the institutional level.

**Individual level:** From an individual level, some teachers do not have personal computers, and they depend on the institution's computers, which are limited compared with the demand. TX5 reported, "I do not have a personal computer; I always prepare my teaching notes using office desktops..." [TX5].

On the same question, TX13 added the following to support the finding, "In my teaching, I prefer the traditional approach. The lack of technological devices and high risk on the use of these technologies are the reasons I can give" [TX13].

These two responses from participants showed some of the teacher educators failed to integrate ICT into their teaching due to a lack of technological devices. However, some reported the high risks associated with using technologies in their teaching and learning process. During the interview, TX11 responded by disclosing that, "My computer collapsed in the mid of the session and had no other means. I postponed the session" [TX11]. These findings reveal the need for training on why and how to use technologies during teaching and learning.

Institutional level: At the same time, the findings show that teacher educators who want to integrate ICT during teaching and learning lack material support from the institutions. Material support includes computers, projectors, internet connection etc. TX4 revealed the following as he said, "...the problem came when the institution did not create the conducive environment for teachers to integrate ICT during teaching. in this case, it includes the availability of enough resources like computers, projectors, internet connection like strong wireless" [TX4].

At the institutional level, TX8 reported, "...but we never have seminars or workshops on how we can transform our teaching to cope with the growth of sciences and technology" [TX8].

The above findings show a lack of support from the management in terms of equipment and professional development. Teacher educators feel they lack enough confidence to integrate technologies into their teaching. Due to severe problems of professional development, TX7 said that, "Some of us using teaching notes of ten years ago; this is no longer teaching but the transfer of knowledge or materials" [TX7].

The findings show that most teacher educators cannot use technologies to sustain their regular teaching, such as accessing teaching materials. Thus, the current status of ICT in many institutions affects future student-teachers who lack mentorship during their teachers' training to improve their future classroom teaching.

These findings show those who have a positive attitude toward integrating technology during the teaching and learning process lack motivation or support from the management and fellow academic staff. One respondent admitted, "There is no motivation and support from the management at all. We have a quality assurance department in our institution, but their assessment is on whether the lecturer has taught or not. Not on the approaches used during the teaching and learning process" [TX9].

This situation affected many teachers who needed motivation and support during the integration process. The participants suggested increasing motivation for those trying to integrate technology into their teaching. At the same time, the department responsible for quality assurance should assess the methodological approach to enhance teacher educators to improve their practices.

### 4.1.6. Disregarding Pedagogical Knowledge

Some respondents perceived that having ICT competencies is enough for them to integrate ICT without having pedagogical knowledge (PK). During the interview, one respondent said, "...I know how to use a computer, so can integrate ICT in my teaching" [TX6], while others admitted that pedagogical knowledge (PK) and ICT competencies are essential for integrating ICT into education. In contrast, some respondents perceived ICT competence and pedagogical knowledge to be complementary. At the same time, others perceived pedagogical knowledge as a prerequisite, so a person can learn how to integrate ICT into education. It should be clear that pedagogical knowledge is a crucial component of ICT integration.

*4.2. Strategies of ICT Integration in Teaching and Learning*

The second objective examined how teacher educators integrate ICT into teaching and learning. The findings revealed various strategies that range from individual teachers to institutional support. These strategies are using ICT in education, using social media, assisting learners in searching and downloading materials from credible sources, and using college journal subscriptions. The results of in-depth interviews with selected academic staff showed that some integrated ICT in their teaching and learning. The following themes represent their practices of ICT integration.

4.2.1. The Use of ICT in Education

We categorized the findings on the use of ICT in education into two groups: (i) the use of software or tools applications and (ii) learning through ICT or ICT use in learning.

The use of software or tools applications: This category involves using available computer software to effectively accomplish given tasks. Effective use includes using Microsoft word, Excel, PowerPoint, or whiteboards when explaining the science model. With these findings, teacher educators' practices vary with their experiences when using computer software. During an in-depth interview, some of the respondents disclosed the following:

> *"I always prepare PPT before the session, but I can't insert moving clip art or animation for catching learners' attention".* [TX14]

> *"I can use Microsoft word more effectively than other computer software".* [TX11]

> *"I can use the computer software of my choice; the biggest problem is... sometimes I fail to integrate into the teaching process".* [TX7]

> *"Computer applications are not so difficult to use; the problem is how to make learning funny and interesting".* [TX5]

The above responses from the participants show that some teacher educators can use computer applications or software in their teaching and learning process. They highlighted their practices and challenges during the integration. However, the findings proved there were insufficient skills or ability to integrate into the education context.

Learning through ICT or ICT use in learning: The findings divided learning with ICT or ICT use in learning into three main categories: (i) Computer-Assisted Learning (CAL), (ii) Computer-Assisted Inquiry (CAI), and (iii) Distance Learning Approach (DLA).

Firstly, Computer-Assisted Learning (CAL) is highlighted as the interaction between teachers or students with a computer system designed to assist them in the learning process. Teachers or students can interact with a computer system or education software or tutorials for their learning. In this regard, the findings show that some teacher educators use computers to record different tutorials, which helps students learn. The following respondents highlighted:

> *"I do record videos for practical sessions and send to my students, this helps the learner to understand the subject matter, and they can revise, playback anytime they want".* [TX3]

> *"This approach makes learning interesting, cooperative and avoids cumbersome".* [TX11]

> *"I did experiment a few years ago when teaching biology course, the class assisted with tutorials performed better compared to the class not assisted with tutorials".* [TX9]

Secondly, Computer-Assisted Inquiry (CAI) involves using ICT as an aid or tool for collecting information or data from various sources for supporting scientific reasoning and research. This approach uses technology as an agent for interacting with a written piece of information or data through the internet. In this section, the participants highlighted the following:

> *"My responsibility is to guide students in verifying their information or collected data through the internet".* [TX3]

*"I can use SPSS and other software for data analysis. The little knowledge helps to derive the feasible solution of the problem addressed".* [TX7]

*"Through this approach, help us to compare and contrast the piece of information from the field".* [TX11]

Thirdly, in the Distance Learning Approach (DLA) teachers receive information or send information to the students or receive it from various sources, for example, electronic email, chat rooms, videoconferencing, and Wiki. Teachers educators had the following replies:

*"I can receive and send emails to my students; the approach is easy for giving feedback to my students".* [TX6]

*"To me, the official account for feedback is email. It is not complicated compared to other accounts".* [TX1]

*"Sometimes, we use Wiki for learning and feedback, some teachers like chat room".* [TX5]

*"I believe feedback or information is essential during the learning process, and I am flexible to learn and use any account for information and feedback to my students"* [TX2]

### 4.2.2. The Use of Social Media

The findings showed that some teacher educators used social media to enhance their teaching and learning process. These social media include social networking such as WhatsApp, Facebook, Instagram, WeChat, and YouTube. The researchers started to identify those who used social media in their teaching from the faculty members. Among the mentioned social media applications identified, the most used was WhatsApp. During the in-depth interview, participants reported the following:

*"I prefer to use WhatsApp compared to other applications, due to simplicity and convenient".* [TX11]

*"I can post the classroom session, reading materials and other important information to my students through this Application".* [TX9]

*"I have my YouTube channel; students are encouraged to access some of the reading materials and tutorials through my channel. This approach is effective to me since the teaching is extended from the classroom setting to outside the class"* [TX11]

Other respondents highlighted the difficulties of the approach, such as time for preparation, identifying content which complies with the lesson, and challenges in creating, posting, rating, sharing, and commenting.

### 4.2.3. Assisting Learners in Accessing and Verifying Useful Information

Among many other approaches, the findings show that teacher educators assist learners in selecting materials from the internet. The assistance is associated with accessing bulky information online and verifying whether it is valuable in learning or discarding them. Learners feel confused and fail to select the required learning materials. Some of the participants highlighted:

*"Well . . . my task is to help students learn, through the selection of reading materials and giving the guidance on how to find and testing the validity".* [TX6]

*"When students collect their work, I always request them to submit the materials used and other sources of information".* [TX2]

Inadequacy of facilities has negatively affected the accessibility of learning materials for teaching and learning. Other respondents reported limited or inadequate technological resources and inadequate internet connection. Based on the above arguments, teacher educators' roles have been extended and include helping learners in the gathering of information, and evaluating and downloading teaching and learning materials.

#### 4.2.4. Teacher Educators' Use of College Journal Subscriptions

This theme aimed to find whether the college allows academic staff to access different subscribed journals. The opportunity includes free access to and download of articles, books, and other published documents relevant to the education field. Free access to additional materials from relevant journals helps teacher educators be updated and understand various issues around their profession. Regarding the in-depth interview conducted, TX11 reported that, "yes...we can access some journals and have access of free downloads", and TX14 also revealed that, "I can download some articles from big journals free of charge, thanks to the college for subscription".

The findings concurred with the observation that the subscribed journals from reputable databases were accessible through the online library that was open to all staff.

From the above feedback, the free access provides teacher educators with awareness of issues in a broad context and disciplinary teaching and learning scholarship. Furthermore, teachers can explore teaching by carrying out action research. The research is valuable and applicable to solving education problems.

### 5. Discussion

The study focused on understanding the perceptions and practices of teacher educators integrating ICT in their teaching. The findings were summarized in the model, as shown in Figure 1. This section provides a discussion of the findings in three main categories: (1) findings related to their perceptions, (2) findings related to practices, and (3) the interplay between perceptions and practices on integrating ICT in teaching and learning.

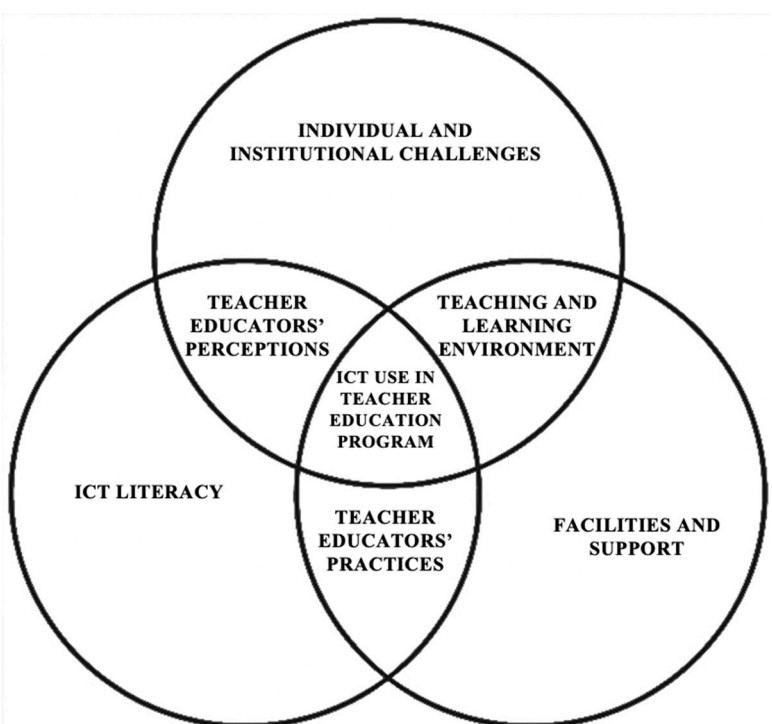

**Figure 1.** Constructs of teacher educators' perceptions and practices on ICT use.

#### 5.1. The Censorious Factors Linked with Teacher Educators' Perceptions of ICT Use

The findings indicated that most teacher educators do not understand the logic behind using ICT in their teaching and learning process. In contrast, others perceive it negatively, despite the existence of college and national policies guiding the usage. Several studies found the same, in which teachers do not understand the significance of technology in teaching, despite being in a teaching environment equipped with ICT facilities [22,52,53]. Their studies suggested infusing technology into the entire teacher education programs so

that teacher educators and preservice teachers can understand the reasons behind using technology. These findings are similar to [54,55], which showed the necessity for full integration of technology throughout the curriculum to provide teachers and students with competencies on ICT use. The findings imply that some teacher educators are unaware of the ICT policy for Basic Education in Tanzania that advocates the integration of ICT in all basic education levels as a content and a pedagogy to facilitate smooth learning. Otherwise, this negatively affects the preservice teachers who need mentorship from their teachers. Without full integration, their skills and knowledge become isolated and unused [25,56]. At the same time, the UTAUT theory suggests teacher educators need to perceive positive ICT use in their teaching, enhancing their motivation for using technology.

Similarly, the study showed some participants narrowly understood technology equipment, while others suffered from computer anxiety. These findings correspond to Pamuk et al. [56] who found that teachers lack enough skills to use technology devices in their teaching, which prevents them from using effective educational technologies. The adopted theory highlighted that inadequate skills in the use of technology can be associated with inexperience, and suggested professional training on how to use new technologies in the teaching and learning process. However, the sophistication and frequent updating of ICT programs and facilities leave some technology adopters behind. Professional training will help teachers improve their practices and experience and overcome their anxiety about technology use. However, Hennessy [1] proposed teacher educators cooperate with other educational experts to develop a working model of technology at the micro level.

Moreover, this study showed teacher educators disregard pedagogical knowledge as an aspect of technology due to contextual factors like short instructional time, large classes, and technical challenges in setting ICT for lesson delivery. While Koehler [57] identified that technology could not be separated or considered unrelated from contexts and teaching tasks, the findings reveal that some teacher educators do not prepare for their sessions in advance, claiming that technology is a challenge. At the same time, technology cannot detach itself from the subject matter and pedagogical knowledge. Teacher educators should consider technology without separating subject matter and pedagogical knowledge when teaching and learning.

### 5.2. The Critical Factors Related to Practices of Teacher Educators on ICT Integration

Discussing the practices of teacher educators on ICT use in their teaching should be seen as the basis for technology integration in their instructional practices [2,58]. The findings show that integrating ICT in the teaching and learning process to some teacher educators simplifies the whole teaching process, while to others, it complicates the process. As teachers use technology to deliver the teaching, this is similar to [22,59] which found that there was an improvement in science courses when using technology. This improvement is associated with using technology in experiments that need practical assistance from videos and natural objects.

This impact also increases students' motivation and engagement with the lesson during teaching and learning [25]. However, these findings are contrary to Cacheiro-Gonzalez et al. [60] who found a low impact on the students' learning when using technology in physics experiments in remedial sessions. The study asserted that there is a need to inform the students about the goals of a particular experiment; otherwise, the integration of technology will not assist students in understanding the subject matter. For that reason, teachers' technology competencies can improve the quality of lesson delivery.

Besides the technology used in science learning, the findings showed some participants used social media such as WhatsApp and showed a narrow understanding of other learning platforms such as Zoom meeting, E-learning, etc. These results are similar to [56,60] which studied whether these online platforms showed significance in student learning. Their studies showed learning platforms promote autonomous learning (86.5%), followed by a provision of course materials (81.1%), and students' engagement (78.4%). The findings are also similar to Kurucay et al. [61]. They found it was the easiest way to

manage the essential elements of the course, such as content, activities, and evaluation, as well as amplifying communication with peers. With this fact, [24,52] proposed that universities and colleges should continue to advance their digital resources repositories and learning platforms by improving the reliability, accessibility, and user-friendliness of learning management systems.

Furthermore, the findings showed that participants can access learning materials through journal subscriptions and by helping preservice teachers gather, evaluate, and download learning materials. Their teachers' assistance allows them to use appropriate materials for their learning. However, teachers' ability to access learning materials from different journals helps to increase students' engagement, enhance collaboration, and effectively find materials to support students' learning. This finding is analogous to Dahlstrom [62] who proposed the extension of teachers' roles in this digital age due to excessive learning materials on the internet.

### 5.3. The Interplay between Perceptions and Practices on Integrating ICT

Identifying the relationship between teachers' perceptions and the level of technology integration is essential for predicting the use of technology among teacher educators [55,63]. However, their internal beliefs and perceptions about technology integration are sometimes more challenging to understand and overcome because they are subjective [33]. The findings show that those with positive perceptions of integrating technology intend to learn or use technology in their teaching. At the same time, those with negative perceptions showed less intention to learn using technology. These findings concur with Darling-Hammond [64], who found teachers with positive perceptions showed a high level of technology adoption and enhanced their competencies in critical thinking, problem-solving, and promoting collaboration with peers.

The study by Sawyer [63] reflects this finding, as it found that perceptions and technology use were statistically significant. As the perceptions increase, the level of technology use also significantly increases. Thus, Kim et al. [65] highlighted teachers' private theories about technology use relate to their personal beliefs about teaching and technology integration practices. The findings contradict Zervas et al. [31], who found a disconnection between the stated goals of technology integration and the actual practices of integrating technology in the classroom. However, this does not imply causation. However, in this study, those with positive perceptions showed high interest in learning or using technology in their teaching. Due to the methodological limitation of a small sample, the current study does not provide enough evidence to conclude that perceptions and practices were correlated but builds the foundations for further studies to investigate the existing relationships.

### 6. Conclusions

The study explored the perceptions and practices of teacher educators on ICT integration in teacher education programs. Specifically, the study intended to understand teacher educators' perceptions and the approaches to integrating ICT in the teaching process. Concerning perceptions, the study has found that most experienced teacher educators lack digital skills as they question the rationale of using ICT in teaching. Young and inexperienced teacher educators show readiness in using ICT, but inadequate facilities, time shortage per period, and the vast teaching load were the main challenges. Again, the study uncovered various approaches to integrating ICT in teaching preservice teachers. Teacher educators use different softwares as teaching and learning platforms. Sometimes, they use social media and subscribe to various academic journals. Moreover, teacher educators assist learners in surfing, searching materials, and evaluating them for reliable and valid information that constitutes their learning.

### 7. Implications

The findings indicate that updating teacher educators' digital skills is essential. Enhancing digital literacy assures the continuous use of ICT in teaching and learning. The

technological revolution has seen many ICT-related programs, software, and applications that require them to be oriented to enhance their literacy and confidence in using them in teaching. The findings show that teacher educators with more years of working experience are less motivated to use ICT. Again, a conducive environment for integrating ICT in teaching requires teacher educators to have a manageable workload and supportive facilities.

Adopting modern approaches to integrating ICT into teaching and learning is necessary. The study revealed the use of software and other applications. The world is now moving to artificial intelligence as they are getting rid of the old technologies in education. Researchers consider social media the most convenient, as most teacher educators and students can access them. These social media have the possibility of enabling synchronous and asynchronous modes of teaching and learning. Since ICT integration in teaching and learning allows learners to access vast materials online, it is vital to enhance a mechanism to ensure the validity and reliability of the materials. The skillful selection of appropriate materials reduce the chances of accessing counterfeit materials that might jeopardize the quality of learning.

**Author Contributions:** Conceptualization, A.I.N. and G.S.; methodology, J.E.K., A.I.N. and G.S.; writing—original draft preparation, A.I.N. and J.E.K. writing—review and editing, J.E.K., G.S. and A.I.N.; visualization, G.S.; supervision, G.S.; project administration, G.S.; funding acquisition, G.S. All authors have read and agreed to the published version of the manuscript.

**Funding:** This research was funded by ICER202001, International Joint Research Project of Huiyan Internatinoal College, Faculty of Education, Beijing Normal University.

**Institutional Review Board Statement:** The study was conducted in accordance with the Declaration of Ministry of Education, and approved by University of Dar es Salaam (code CD5/12E) (05//2020).

**Informed Consent Statement:** Informed consent was obtained from all subjects involved in the study.

**Data Availability Statement:** Data will be available upon request.

**Conflicts of Interest:** The authors declare no conflict of interest.

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
