# Peer review of "Understanding Teacher Educators’ Perceptions and Practices about ICT Integration in Teacher Education Program"

_education, doi:10.3390/educsci12080549_

Round 1
Reviewer 1 Report
There are significant English language issues that make reading to understand the ideas presented difficult. The word choice is not reflective of standard English, rather words that tangentially fit the meaning being expressed are used. For example in line 101-102 " which intended at concocting learners prepared ..." is not an accurate way to express the idea, "the plans were created to prepared learners for the 21st century ..."
The citation numbering system does not full replace the authors' name in the text of the paper. For example in line 84, it is not correct to say "[19] highlighted the calmest of technology useful can be leisurely through users' feedback." The name of the author followed by [19] is needed, and the remainder of the sentence does not make any sense in terms of word choice and grammatical structure.
I did not find the figure representing the findings adequate. As I read this diagram, it suggests to me the faculty who have overcome the barriers use ITC in the ways shown in red. But the text suggests to me, the barriers belong to different individuals than the implementation. It is not the information that is the problem it is the diagrammatic issue selected to represent them. I don't see an embedded relationship between stated beliefs and practices - they are separate categories
Author Response
Kindly receive the feedback.

Reviewer 2 Report
Regarding the previous review, they have introduced the requested improvements as far as possibleIn general, a significant improvement is observed in the article. Specifically:
- Regarding the methodological aspects, the authors have adjusted the methodology section, the design is phenomenologic, it a substantial improvement is observed when adjusting them to at this design.
- Regarding validation indicated in the previous review , the description of the validation of the instruments has been given and additional methods have also been included to ensure data triangulation. They have clarified better how the instrument used was validated .
- Regarding the content of the interview, the revised draft has included the list of key interview questions that were used in the data collection that allows a better understanding of its.
Author Response
Thanks for your positive feedback.
This manuscript is a resubmission of an earlier submission. The following is a list of the peer review reports and author responses from that submission.
Round 1
Reviewer 1 Report
The study explored the perceptions and practices of educators about ICT integration in teacher education programs.
Introduction and review of the literature
It is written that- "This is not limited to internet accessibility in the institution and the knowledge of teacher educators on how to use the existing technologies for the advancement of the preservice teacher." It is advisable to write what is required beyond that according to the literature. There is a broad reference dealing with systemic support, school culture support and customized training programs.
The Methodology part
The methodological description is detailed and clear
The issue of digital natives is presented as a myth in many studies. This is because mastering the technology by itself does not indicate a wise use and does not necessarily show proper cognitive abilities while using technology. This reference should be addressed in the context of this study.
You wrote "teachers sticking to traditional teaching methods". However, technology is not pedagogy. Using technology in education can strengthen traditional teaching or strengthen innovative pedagogy. This is in accordance with the instructional design of the teaching/learning process. Treating technology as innovative by itself is outdated and irrelevant.
The findings
The findings look like they were taken 20 or 30 years ago. There is nothing new in them, neither in the context of the digital natives compared to digital immigrants, nor in the context of what it is the pedagogical knowledge required to carry out technology in the teaching and learning processes.
Beyond that, emphasizing computer use capabilities and the accessibility of digital equipment from home is irrelevant in many parts of the Western world.
Reviewer 2 Report
Once the manuscript has been reviewed:
It´s a worthwhile study, this project could be accept after very minor revision.
Communication is very relevant when addressing a topic of great interest in teacher training, it is current and timely due to the importance that the use of ICT has gained with the pandemic and the unexpected interruptions of face-to-face classes.
The paper is well structured and adequately establishes the theoretical framework and the objective. The results collected are well written and clear, it could be interesting if, in addition to narrating them, graphic representations of the information obtained had been made to visualize the results better.
The methodology used is sufficiently explained and adequate information is given on the processes carried out to obtain and organize the data.
Regarding the methodological aspects, the use of semi-structured interviews and the use of qualitative methodology stand out. It should be clarified if the instrument used was validated in some previous way or if it comes from a previous work of the team itself or of other teams, it is not very clear.
It would be important to know better the content of the semi-structured interview, how many questions, how many questions per category, relationship of the questions with the established categories, ... It is recommended to define the categories (section 4.1), although when reading the results, they are sufficiently delimited.
It should, since the data was disaggregated by sex, refer to results that show the different (or equal) perceptions according to gender. It might be of interest to provide clearer data on the differences in perceptions between teachers in different areas or departments, although there is a specific reference to science teachers.
It is proposed to avoid the use of the term "new technologies" (line 145) for the time that they have already been present in our society.
Reviewer 3 Report
I found the topic interesting and believe their are some interesting findings about the faculties' beliefs and use of ICT. However there are some substantial issues that need to be addressed.
1. the author/s should work with a native or near native English language speaker on the translation of their paper. As the manuscript is now, the English usage makes it close to unreadable.
2. The paper is methodologically listed as a case study, however as it is constructed in this writing, it is not a case study. There is no discussion about how the ideas of the individual participants come together to tell us about the institution (case). The entire study methodology should be framed as something other than a case study, possibly a phenomenological study, or simply a qualitative study. The author/s do not build a case or even discuss the institution where the participants are all instructors. Even if this is not a case study, more information about the location of the research is needed - country, kind of institutition, kind of curriculum used and other details are needed for the reader to understand the how to interpret the provided responses.
3. The description of what is required for qualitative analysis is not needed, but more detail and examples of how the researchers carried out the phases of the research would support their conclusions. They author/s include no information about what codes were used or how themes were derived from codes. Sample codes and themes, and how they were derived from the theoretical position of the research are needed.
4. The author/s suggest there is information about the interview questions asked, but I could not find this. It would be helpful to have some example questions that helped them arrive at the their themes, so I can understand if the interview questions gave the interviewees the opportunity to openly discuss their ideas, as is needed in qualitative research.
5. I was unclear about why the authors choose to review the use of ICT in the global context and how these global situations are relevant to understanding the research questions and the specifics of the case study. Since we do not know the context of the study it is difficult to find the fit. The discussion of the case study needs to be more concrete and clearer. State, that a single institution was the case, the data was gathered from faculty and instructors through interviews. In my experience, in case studies multiple sources and kinds of data are gathered, so simply interviewing multiple people would seem to provide a lack of data sources for triangulating ideas is used. Were there observations of available resources or of teaching, artifacts gathered from the faculty. It seems as if the entire case is built on self-report.
Other more detailed items would include:
the need to define all acronyms (ICT, TPD) and provide examples of what, as author/s you are including in these definitions, either descriptions or examples. I understand what these are in my countries context but because the author/s are discussing global issues more guides are needed.
Tables 1 & 2 can be combined as the only – take out table 1, and add in the degrees held with their positions. It is not clear why the information on the duration of the interview is needed.
There are numbering and formatting issues in the presentation of the data. Different styles are used, which is confusing.
As I mentioned at the start, the findings and ideas of the participants were interesting and the discussion and implication interesting and worthy of sharing.